# Analyzing online public commentary responding to the announcement of deemed consent organ donation legislation in the Canadian province of Nova Scotia

**Alessandro R. Marcon**[1]*, **Darren N. Wagner**[1], **Christen Rachul**[2], **Matthew J. Weiss**[3,4,5]

**1** Health Law Institute, Faculty of Law, University of Alberta, Edmonton, AB, Canada, **2** Rady Faculty of Health Sciences, University of Manitoba, Winnipeg, MN, Canada, **3** FRCPC, Transplant Québec, Montréal, QC, Canada, **4** CHU de Québec, Université Laval Research Centre, Population Health and Optimal Health Practices Research Unit, Trauma-Emergency-Critical Care Medicine, Université Laval, Québec, QC, Canada, **5** Canadian Donation and Transplantation Research Program, Edmonton, AB, Canada

* marcon@ualberta.ca

**Data Availability Statement:** A minimal anonymized data set is available here: 10.6084/m9.figshare.19189541.

## Abstract

### Background

The Canadian province of Nova Scotia recently became the first jurisdiction in North America to pass deemed consent organ donation legislation. The announcement of this legislation generated substantial online discussion, which we analyzed to provide insights on public perception.

### Methods

We performed directed content analysis on 2663 user-generated comments appearing on two widely-shared Canadian Broadcasting Company (CBC) articles published online in April 2019. We determined levels of support and opposition in comments and described the specific rhetoric used for doing so. We also performed one-way ANOVA and Pearson chi-square tests to determine how the comments were being received and engaged by other users.

### Results

A range of commentary was present in both support and opposition to the changes in legislation. There were more negative than positive comments, and negative commentary generated more replies. Positive comments were received more positively by other users while negative comments were received more negatively. The total sum of negative comments was greatly influenced by a small number of very active participants. Negative commentary focused more on broad concepts and principles related to government, power, and individual rights rather than specific issues in the Nova Scotian context. Substantial issues of trust in the government and healthcare system were evident.

**Funding:** The authors thank and acknowledge Health Canada, Genome Canada, Genome Alberta, and the Canadian Institutes for Health Research for their generous support of Legislative Strategies to Improve Deceased Organ Donation in Canada: A Special Focus on Evaluating the Impact of Opt-Out Legislation in Canada LEADDeR and Precision Medicine CanPREVENT AMR: Applying Precision Medicine Technologies in Canada to Prevent Antibody Mediated Rejection and Premature Kidney Transplant Loss.

**Competing interests:** The authors have declared that no competing interests exist.

## Conclusions

There were strong positive and negative sentiments expressed in the comments, but the total sum of negativity in the comments was significantly influenced by a small number of commentators. Analysis on the presumed consent concerns can be helpful to inform public outreach efforts.

## Introduction

In 2019 the Canadian province of Nova Scotia became the first jurisdiction in North America to pass deemed consent organ donation legislation [1] as a means of addressing low organ donation rates [2]. In 2019, Atlantic Canada had lower rates of donors per million population (13) compared to other Canadian provinces (Canada average of 21) [2], while internationally, Canada's organ donation rates were lower than many other national averages [3]. Other provinces have previously considered implementing presumed consent donation legislation, but criticisms prevailed regarding the perceived public reaction and the extent to which presumed consent laws would increase organ donation rates [4–6].

Research shows that this legislation is not, in and of itself, a sure remedy for organ donation shortages [2, 7, 8]. A successful donation and transplantation system relies on multiple factors, such as a well-functioning national registry, a fair and equitable organ allocation system, and donation laws that align with the public's social and ethical bearing [5, 8–10]. In the past, some countries, such as Singapore, Brazil, and Chile [11–13] implemented presumed consent legislation but were unsuccessful at increasing donation and transplantation rates. England and the Netherlands, however, have seen increases in donation and transplantation rates while transitioning to presumed consent systems [14]. The efficiency of any donation consent model relates to several important considerations, such as accommodating next-of-kin and educating the public on donation and transplantation measures [15, 16].

Following the passing of legislation in Nova Scotia, the Canadian Broadcasting Company (CBC) published an article on its website on April 2nd, 2019 announcing the changes, "Nova Scotia to become 1st in North America with presumed consent for organ donation" (Article 1). Two days later, the CBC published a follow-up article titled, "Nova Scotia's opt-out organ donation move sparks mixed reaction" (Article 2). These two articles were shared widely on Facebook and Twitter, and generated substantial online discussion with more than 2,500 comments and replies. CBC News Online is one of the most used news brands in Canada [17] and in February 2022, www.cbc.ca was the only news website listed in the 50 most visited websites in Canada [18]. Research has demonstrated how user-generated comments influence audiences, finding that comments, and interactions with comments (e.g., "likes" and "dislikes") not only influence readers' perceptions of public opinion but also their evaluations of the news coverage [19–21]. Because the Canadian public is increasingly using online sources to investigate and debate health-related topics [22], online discussions about pressing health topics can provide valuable insight into public perceptions. In this research project we analyzed the comments generated by these two highly-shared articles on the deemed consent legislative changes in Nova Scotia. We hypothesized the existence of positive and negative opinions that may potentially have influenced public perception and debate on this topic. Our analysis determined the characteristics of that commentary, and how other online users interacted with those comments.

## Methods

### Data collection

On May 24[th], 2019, we downloaded all user comments and replies (replies to comments and replies to replies) posted to the two CBC articles published on April 2[nd] and April 4[th], 2019 using a program designed to interact with the CBC's online application program interface. We therefore downloaded all comments which had been left in discussions, and not removed by moderators, between the publication dates and May 24[th]. Our program, the complete script of which is available here: https://github.com/philbot9/python-cbc-comment-scraper, captured and organized all comments and corresponding replies in a sequential order, while also capturing the usernames of all participants and the number of likes and dislikes given to each comment and reply. The complete data sets were organized in CSV files. The CBC, a public broadcaster funded by the Canadian government, states the following with respect to online commenting, "The information gathered ensures transparency in public debate and accountability of participants. Information may also be used for statistical and informational analyses and public opinion trend monitoring, among other things" [23].User comments consist of publicly shared media content, and ethics approval was therefore not required. Nonetheless, no effort was made by the research team to link user names to actual identities or demographic data, and no usernames appear in this report to help protect privacy.

### Data analysis

We performed a directed content analysis [24] mixed with a descriptive qualitative analysis using a "general inductive approach" [25] to analyze whether the comments and replies, in sum, expressed a negative or positive perspective on the new legislation, and what benefits of, or concerns with the new legislation were expressed. Comments supporting and promoting the new legislation as well as donation and transplantation broadly were deemed positive whereas comments critiquing the new legislation as well as donation and transplantation broadly were deemed negative. Comments which did not exhibit negativity or positivity, or which did not present a clearly defined negative or positive position, were coded as neutral. We also coded for participant activity according to output, specifically whether they provided one, two, or more than two comments. All coding analysis was conducted in shared Google Sheets.

   We performed a Kruskal-Wallis test with pairwise post-hoc Dunn test to determine whether negative, neutral, or positive comments were generating more negative, neutral, or positive feedback (based on likes or dislikes), and whether the negative or positive comments were generating more discussions (based on the number of replies). We also performed the Pearson's chi-square test of independence to determine the relationship between how much a participant contributed comments and whether their comments were negative, neutral, or positive. We used SPSS v.27 to perform all statistical tests. We did not perform any demographic or metadata analysis on the participants' accounts, aside from counting the number of contributions made.

   Given that presumed consent legislation had already passed, we placed focus on the negative responses in order to help inform public outreach efforts. As such, we performed directed content analysis on all individual negative comments and each reply thread (the sum of negative replies to a positive or negative comment), quantifying the specific types of concerns expressed and particular trends in rhetoric. The two coders developed a categorization of concerns when performing the initial determination of positive, negative, or neutral comments, which ultimately consisted of 18 items, solidified through a consensus reaching session (S1

Text for the complete categorization list). Coding for each concern on the list was not mutually exclusive. To analyze the positive and neutral comments, we performed a general qualitative analysis using a "general inductive approach" [25], whereby each of the two coders made notes on the comments and replies left in the discussions. Positive and neutral comments were not quantified with regards to specific arguments or sentiments. Following the coding of all comments and replies, a collaborative session between the coders was held to ensure coding consistency and to reach consensus on the most salient topics and themes.

## Findings

**Content analysis summary.**   There were a total of 2663 comments and replies to the two articles, consisting of 629 comments and 2034 replies, posted by a total of 512 total unique participants. The ratio of comments to replies was nearly identical in Article 1 (1: 3.25) and Article 2 (1: 3.18). Some participants had contributed to the discussions in both articles (n = 54, 10.6%), but most had contributed to only one (n = 458, 89.5%) (Table 1).

Both articles yielded more negative than positive comments. Of the 629 comments, 339 (53.9%) were negative, 172 (27.3%) were positive, and 118 (18.8%) were neutral (Table 1). Article 1 demonstrated a higher percentage of negativity in comments ($n = 251$, 56.0%) than Article 2 ($n = 88$, 48.6%), but the difference in tone between the two articles was not significant, ($X^2 = 3.08$, $df = 2$, $p = .21$). There was a significant difference between the number of replies to comments based on the tone of the comment (Kruskal-Wallis, $X^2 = 22.426$, $df = 2$, $p = < .001$). A pairwise post-hoc Dunn test with Bonferroni adjustments was only significant for negative vs. positive comments ($p = < .0001$) and negative vs. neutral comments ($p = .007$).

The two articles' comments generated a total of 8344 reactions (likes and dislikes), consisting of 4529 likes (54.3%) and 3815 dislikes (45.7%). Negative comments, in sum, received a higher percentage of dislikes (53.3%) than likes (46.7%). In contrast, 65.0% ($n = 1824$) of the reactions for positive comments were likes, and 57.9% ($n = 609$) of the reactions for neutral comments were likes (Table 1). There was a significant difference between how many likes each comment received depending on whether the comment was negative, positive, or neutral (Kruskal-Wallis, $X^2 = 84.291$, $df = 2$, $p = < .0001$). A pairwise post-hoc Dunn test with Bonferroni adjustments was significant for positive vs. negative comments ($p = < .0001$), positive vs. neutral comments ($p = < .0001$) and negative vs. neutral comments ($p = .016$). There was also a significant difference between how many dislikes each comment received depending on whether the comment was negative, positive, or neutral, (Kruskal-Wallis, $X^2 = 56.414$, $df = 2$, $p = < .001$). A pairwise post-hoc Dunn test with Bonferroni adjustments was significant for positive vs. negative comments ($p = .007$), positive vs. neutral comments ($p = < .0001$) and negative vs. neutral comments ($p = < .0001$). Thus, positive comments typically generated more positive reactions and negative comments received more negative reactions.

Of the 512 unique participants, 425 contributed negative, positive, and neutral comments (see S2 Table for complete numbers). There were more participants contributing negative comments ($n = 184$) than there were contributing positive ($n = 146$) or neutral comments ($n = 95$). Those contributing negative comments, however, were considerably more active. In the two articles, the 339 negative comments were made by 184 participants for an average of 1.84 comments per participant, compared with 1.18 for positive comment contributors, and 1.24 for neutral comment contributors. Further, there was a significant difference in whether a participant left one, two, or more than two comments and whether their comments were negative, positive, or neutral, ($X^2 = 60.37$, $df = 4$, $p = < 0.001$). That is, of the 226 comments that were provided by participants who contributed more than twice, 71.7% ($n = 162$) were negative comments. In comparison, only 132 (43.9%) of 301 comments left by participants who

**Table 1. Total data summary for number of participants, comments, replies, and reactions in the online commentary generated by two CBC articles on the topic of presumed consent in Nova Scotia.**

| Content analysis category | Article 1 | Article 2 | Total (%) | Average |
|---|---|---|---|---|
| **Total unique participants** | 381 | 185 | 566 | |
| Participants in both articles | | | 54 (10.6%) | |
| Participants in only one article | 327 | 131 | 458 (89.4%) | |
| Unique participants across both articles | | | 512 | |
| **Number of comments** | 448 | 181 | 629 | |
| **Number of replies** | 1458 | 576 | 2034 | |
| **Total comment and replies** | 1906 | 757 | 2663 | |
| **Ratio of comments to replies** | (1: 3.25) | (1: 3.18) | | |
| **Number of reply threads** | 368 | 139 | 507 | |
| **Number of comments** | 448 | 181 | 629 | |
| Negative comments | 251 (56.0%) | 88 (48.6%) | 339 (53.9%) | 52.3% |
| Positive comments | 115 (25.7%) | 57 (31.5%) | 172 (27.3%) | 28.6% |
| Neutral comments | 82 (18.3%) | 36 (19.9%) | 118 (18.8%) | 19.9% |
| **Number of replies…** | 1458 | 576 | 2034 | |
| to negative comments | 971 (66.6%) | 318 (55.2%) | 1289 (63.4%) | 60.9% |
| to positive comments | 285 (19.6%) | 162 (28.1%) | 447 (22.0%) | 23.8% |
| to neutral comments | 202 (13.8%) | 96 (16.7%) | 298 (14.7%) | 15.3% |
| **Number of reactions (likes + dislikes)** | 6179 | 2165 | 8344 | |
| **Number of likes…** | 3289 | 1240 | 4529 | |
| to negative comments | 1647 (50.1%) | 449 (36.2%) | 2096 (46.3%) | |
| to positive comments | 1241 (37.7%) | 583 (47.0%) | 1824 (40.3%) | |
| to neutral comments | 401 (12.2%) | 208 (16.8%) | 609 (13.4%) | |
| **Number of dislikes…** | 2890 | 925 | 3815 | |
| to negative comments | 1861 (64.4%) | 529 (57.2%) | 2390 (62.6%) | |
| to positive comments | 718 (24.8%) | 265 (28.7%) | 983 (25.8%) | |
| to neutral comments | 311 (10.8%) | 131 (14.2%) | 442 (11.6%) | |
| **Total reactions…** | | | | *Per comment* |
| to negative comments | | | 4486 | *13.2* |
| to positive comments | | | 2807 | *16.3* |
| to neutral comments | | | 1051 | *8.9* |
| **Total percentage of likes to dislikes** | | | 54.3% / 45.7% | |
| to negative comments | | | 46.7%/53.3% | |
| to positive comments | | | 65.0% / 35.0% | |
| to neutral comments | | | 57.9% / 42.1% | |

contributed once were negative. The top five participants contributing negative comments represented 2.7% of all negative comment contributors, but their comments accounted for 27.4% of the total negative comments. In contrast, the top five participants contributing positive comments represented 3.4% of all positive comment contributors, but their comments accounted for only 10.3% of the total positive comments (S2 Table).

This trend of increased participation from negative comment contributors was also evident from analyzing the total sum of their comments and replies. For participants who had contributed at least two negative or positive comments, the five most active negative participants contributed a total of 283 comment and replies, accounting for 10.6% of the total comments and replies (*N* = 2663). In comparison, the five most active positive participants contributed a total of 115 comments and replies, accounting for 4.3% of the total (S2 Table).

**Table 2. Negative commentary types and frequency of use for all negative comments and reply threads in each of the two articles.**

| Category* | Negative comments in both articles (n = 339) | | Reply threads in both articles (n = 507) | | Overall Average % | Overall Rank |
|---|---|---|---|---|---|---|
| | Average % across both articles | Rank | Average % across both articles | Rank | | |
| Gov. usurp power | 41.8 | 1 | 23.8 | 1 | 32.8 | 1 |
| Ownership | 12.8 | 3 | 15.5 | 2 | 14.2 | 2 |
| Legal issues | 16.6 | 2 | 8.3 | 5 | 12.4 | 3 |
| "Harvest" | 12.1 | 4 | 8.7 | 3 | 10.4 | 4 |
| Procedures | 7.2 | 9 | 8.5 | 4 | 7.8 | 5 |
| General | 10.6 | 5 | 4.6 | 10 | 7.6 | 6 |
| Liberals | 7.7 | 6 | 4.6 | 9 | 6.2 | 7 |
| Profits | 7.4 | 8 | 3.8 | 13 | 5.6 | 8 |
| Pro-donation | 6.9 | 10 | 3.8 | 12 | 5.4 | 9 |
| Comparisons (of consent) | 7.6 | 7 | 2.8 | 15 | 5.2 | 10 |
| Doctors | 5.4 | 11 | 4.7 | 8 | 5.1 | 11 |
| Improve current | 3.1 | 17 | 5.7 | 6 | 4.4 | 12 |
| Other countries | 4.3 | 14 | 4.1 | 11 | 4.2 | 13 |
| Morally wrong | 5.0 | 12 | 3.3 | 14 | 4.2 | 14 |
| Religions/ Cultures | 2.2 | 18 | 5.6 | 7 | 3.9 | 15 |
| Consultation | 4.8 | 13 | 1.6 | 16 | 3.2 | 16 |
| Infrastructure | 3.5 | 15 | 1.3 | 17 | 2.4 | 17 |
| Dystopia | 3.4 | 16 | 0.5 | 18 | 2.0 | 18 |

## Content analysis of negative comments

The types of concerns and critiques expressed in the total sum of negative comments (*n* = 339) and all reply threads (*n* = 507 (all comments with replies in both Articles 1 and 2)) demonstrated a trend towards critiquing the principle of presumed consent broadly and on overarching principles more so than by detailing specific concerns around implementation procedures and policies (see Table 2 for complete numbers and S3 Table for examples of each type). The broad concerns included issues of government power usurping individual rights (32.8%), ownership of the body (14.2%), and potential legal conflicts (12.4%) regarding, for example, "universal human rights" or rights guaranteed by the Canadian *Charter of Rights and Freedoms*. Some procedural issues were raised (7.8%), such as, around the healthcare system's functional abilities and perceived deceitfulness within the system. Further critiques were raised about the "Liberals," who had passed the legislation (6.2%) and about the "profits" that would be gained from increased organ procurement (5.6%). "Pro donation" criticisms (5.4%) referred to statements by individuals voicing support for organ donation but not for presumed consent. The category of "comparisons" (5.2%) included statements analogizing presumed consent to other practices perceived as unjust, including both hypothetical (e.g., "it's like the government taking possession of your possessions because you're no longer using them") and the business practice of "negative billing" [1]. The idea of physician malpractice (5.1%), expressed the idea that doctors' efforts to keep patients alive would be compromised by the desire to recover organs, either to save someone else's life or to generate profits. Concerns around adequate consultation (3.2%) or infrastructure (2.4%) were scant. For examples of each negative commentary type see S3 Table.

## Themes and topics in positive and neutral comments

The themes and topics in positive and neutral comments determined through the qualitative analysis are summarized in Table 3. As mentioned in the methods, no statistical analysis was

**Table 3. Thematic and topic summary of positive and neutral comments.**

| Summary of positive comments |
| --- |
| ■ Laws do not restrict autonomy–an easy and clear choice to opt out is available |
| ■ Facilitates altruism (saves more lives) |
| ■ Previous opt-in policies have not proved effective |
| ■ Other countries have been successful with presumed consent law implementation |
| ■ Organs go to waste if not used |
| ■ Those who opt-out should not be able to receive an organ |
| ■ Nova Scotians should feel proud of their province for being innovative and progressive |
| ■ Frustration, anger and annoyance expressed as those voicing opposition |

| Summary of neutral comments |
| --- |
| ■General reflections on the discussions (surprise, dismay at others' comments) |
| ■ General political observations |
| ■ Attempts at humour |
| ■ Questions (without apparent rhetorical objectives) on topics such as determining death, family veto, the rationale of those in opposition, organ trade between provinces, accommodating those with mental disabilities |
| ■ General reflections on Nova Scotia and its healthcare, other provinces, different religions/theologies, organ donation science |

performed on these comments. Most notably, positive comments argued that the new laws maintained autonomy and choice, and would ultimately save more lives. Positive comments demonstrated a pride in the province's decision and expressed a desire for other provinces to follow. Positive commentary exhibited frustration, annoyance, and sometimes anger towards those voicing opposition. Neutral commentary typically reflected on the discussions taking place, asked various questions around the new policy, and provided general comments on healthcare, the province, and other jurisdictions in Canada.

## Discussion

Our analysis is the first to examine the robust online commentary regarding reactions to the announcement of Nova Scotia's deemed consent organ donation law. We found a variety of reactions both in opposition to and support of the new legislation. Notably, we observed that the negative commentary was greatly influenced by small group of dedicated commentators, whose concerns were primarily focused on broad principles of individualism and power rather than concerns specific to the Nova Scotian context. As the positive commentary received more likes and fewer dislikes by other readers, our findings–despite not being generalizable to the public–suggest that the amount of negativity is not representative of the total sum of the articles' participants.

  The online public commentary accompanying the two CBC articles that discussed presumed consent legislation for organ donation in Nova Scotia showed a range of perspectives supporting and opposing the new law. Those in favour saw the changes as a means of addressing organ shortages while maintaining individual choice. Positive comments portrayed the law as altruistic and a point of pride for the province being the first in Canada to enact such measures. Those in opposition saw the law as an overreach of government authority, thereby unjustly impinging on the freedoms and autonomy of individuals. Posts expressed sentiments of condemnation on various grounds, including legal issues, ethical impropriety, implementation problems, infrastructure limitations, and insufficient consultation. Some arguments took place among participants debating the issues and merits of the legislation. Others used the platform to ask questions and seek clarification around specific aspects of the law. While the sample size (512 unique participants) is not generalizable to the general public, there are a few important conclusions and practical observations that can be drawn from this analysis.

We observed considerably more negative than positive comments posted in the discussions. This is perhaps unsurprising, given that research has shown how negativity can proliferate online [26] spurred on by the "negativity bias" [27] which makes readers more attracted to and influenced by negative discourse [28]. Initially, the high prevalence of negativity in the discussions gave the impression of a divided public, with many reacting critically to the new legislation. But our analysis showed that contributors of negative comments were, on average, considerably more active than those contributing positive comments (1.84 versus 1.18, respectively), and that negative commentary was produced by a small cluster of more active users who tended to post multiple comments. Indeed, the five most active contributors of negative comments represented 2.7% of those posting negative comments but their output accounted for over 27% of all negative comments. In comparison, the five most active contributors of positive comments represented 3.4% of those posting positive comments but their output accounted for only 10.3% of all positive comments. Further, our analysis showed that positive commentary was reflected upon with more positivity while negative commentary received more negative feedback. Therefore, a relatively small pack of very loud voices might have created a false impression of broad social contention on the topic. It was also observed that some of the negative commentary exhibited conspiratorial elements such as doctors "harvesting" organs for profit or personal incentives, or government bodies acting with nefarious intent. In other contexts, research on the spreading of information and misinformation online has shown how integral and influential some active participants can be in propagating particular storylines or creating chaotic messaging [29]. Importantly, it has been observed that since the roll out of the legislation (January 2020), Nova Scotia is experiencing an increase in tissue donations and organ donor referrals, and fewer than 6% of the residents have elected to opt out [30]. It is possible that automated accounts, or "bots," were contributing to the negativity in the discussions, but analyzing user accounts fell outside of the purview of this research. Future research on the topic could make use of developing software to estimate bot activity in such discussions. Future research could also examine public discussions on other social media sites such as Facebook, Reddit, or Twitter to provide additional insights into the discussions taking place.

Although overrepresented by a handful of very active participants, the negative commentary included some useful indicators for assisting public outreach. The issue of trust permeated throughout the critical commentary. Individuals expressed suspicion around government activities broadly but also expressed mistrust towards health care systems and medical practitioners. Research has shown how a lack of trust in health care systems can negatively impact procedures and outcomes [31]. Other issues pertained to the management of health care records, the determination of death, and the nature of consent more broadly. Users also raised concerns about why the changes to legislation were necessary and whether they were justified.

In response, it would seem beneficial for governing bodies to clearly express why changes around consenting for organ donation were needed, how the policy will be enacted (e.g. power of families to veto), and what is hoped can be gained from the policy. It would likely be helpful to have a range of stakeholders disseminate knowledge, such as medical experts, religious leaders, as well as influential members of racialized and/or marginalized communities. Explanations of the organ donation and transplantation process as well as the legal foundation of the legislation might also prove to be useful for consensus building in some contexts.

As research has detailed how presumed consent is not a panacea for low donation and transplantation rates [7, 8], assessing the various ways in which donation and transplantation procedures can be improved are paramount, including an evaluation of the potentially valuable implementation of presumed consent. Part of this evaluation [32] needs to focus on those who opt out and their reasons for doing so. In being the first jurisdiction in North America to

pass presumed consent legislation, the province of Nova Scotia is providing a valuable learning and evaluation opportunity for other provinces in Canada as well as other jurisdictions internationally.

## Supporting information

**S1 Text. Negative comment coding categories.** Negative comment coding categories.
(DOCX)

**S1 Table. The three most liked, disliked, and replied-to comments from each article.** The three most liked, disliked, and replied-to comments from each article.
(DOCX)

**S2 Table. Summary of participants' comments based on negative, positive and neutral comment contribution.** Summary of participants' comments based on negative, positive and neutral comment contribution.
(DOCX)

**S3 Table. Text examples for each negative comment category type.** Text examples for each negative comment category type.
(DOCX)

## Acknowledgments

The authors extend special thanks to Philip Klostermann for assistance in data collection, as well as Robyn Hyde-Lay, Jade Dirk, Stephen Beed, and the entire LEADDR Activity 3 team for their assistance and support of this project.

## Author Contributions

**Conceptualization:** Alessandro R. Marcon.

**Data curation:** Alessandro R. Marcon, Darren N. Wagner.

**Formal analysis:** Alessandro R. Marcon, Darren N. Wagner, Christen Rachul.

**Methodology:** Alessandro R. Marcon, Christen Rachul.

**Supervision:** Alessandro R. Marcon.

**Writing – original draft:** Alessandro R. Marcon, Darren N. Wagner, Christen Rachul, Matthew J. Weiss.

**Writing – review & editing:** Alessandro R. Marcon, Darren N. Wagner, Christen Rachul, Matthew J. Weiss.

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
