## [Decision Letter · Decision Letter 0]

10 Jan 2022

PONE-D-21-22056Online user comments responding to deemed consent organ donor legislation in Nova Scotia: a divided public or a few loud voices?PLOS ONE

Dear Dr. Marcon,

Thank you for submitting your manuscript to PLOS ONE. After careful consideration, we feel that it has merit but does not fully meet PLOS ONE’s publication criteria as it currently stands. Therefore, we invite you to submit a revised version of the manuscript that addresses the points raised during the review process.

Please address all the comments from reviewer 1 and 2 as they are important for the clarity of the work. Also, address the comments by the editor.

We look forward to receiving your revised manuscript.

Kind regards,

Ronaldo Menezes

Academic Editor

PLOS ONE

Journal Requirements:

2. Please consider changing the title so as to meet our title format requirement (https://journals.plos.org/plosone/s/submission-guidelines). In particular, the title should be "Specific, descriptive, concise, and comprehensible to readers outside the field" and in this case it is not informative and specific about your study's findings.

3. In your Methods section, please include additional information about your dataset and ensure that you have included a statement specifying whether the collection method complied with the terms and conditions for the websites from which you have collected data.

4. Thank you for stating the following in the Funding Section of your manuscript:

“The authors thank and acknowledge Health Canada, Genome Canada, Genome Alberta, and the Canadian Institutes for Health Research for their generous support of Legislative Strategies to Improve Deceased Organ Donation in Canada: A Special Focus on Evaluating the Impact of Opt-Out Legislation in Canada LEADDeR and Precision Medicine CanPREVENT AMR: Applying Precision Medicine Technologies in Canada to Prevent Antibody Mediated Rejection and Premature Kidney Transplant Loss.”

We note that you have provided additional information within the Funding Section that is not currently declared in your Funding Statement. Please note that funding information should not appear in other areas of your manuscript. We will only publish funding information present in the Funding Statement section of the online submission form.

“Health Canada (https://www.canada.ca/en/health-canada.html)

Genome Canada (https://www.genomecanada.ca/)

Genome Alberta (https://genomealberta.ca/)

Canadian Institutes for Health Research (https://cihr-irsc.gc.ca/e/193.html)

Projects funded:

Legislative Strategies to Improve Deceased Organ Donation in Canada: A Special Focus on Evaluating the Impact of Opt-Out Legislation in Canada LEADDeR and Precision Medicine CanPREVENT AMR: Applying Precision Medicine Technologies in Canada to Prevent Antibody Mediated Rejection and Premature Kidney Transplant Loss

No, the funders had no role in study design, data collection and analysis, decision to publish, or preparation of the manuscript.”

Additional Editor Comments:

The paper does a statistical analysis of commentary about legislation regarding organ transplantation. The paper is quite simple and it doesn't introduce any new methods. However the topic is quite important and deserves publication. I'd like to point out the following concerns

- The analysis of just one dataset makes the relevance doubtful, as we don't know how bias are the comments.

- Do the comments represent a particular demographic? Does the site represent the population?

- Are robots present? Is there a way to find out from the content that the "language" may not be of a person but maybe automated?

Reviewers' comments:

Reviewer's Responses to Questions

**Comments to the Author**

1. Is the manuscript technically sound, and do the data support the conclusions?

Reviewer #1: Yes

Reviewer #2: Yes

2. Has the statistical analysis been performed appropriately and rigorously? 

Reviewer #1: Yes

Reviewer #2: Yes

3. Have the authors made all data underlying the findings in their manuscript fully available?

Reviewer #1: Yes

Reviewer #2: Yes

4. Is the manuscript presented in an intelligible fashion and written in standard English?

Reviewer #1: Yes

Reviewer #2: Yes

5. Review Comments to the Author

Reviewer #1: This work attempts to understand public perceptions regarding organ donation which is an increasingly relevant issue in public health, specially its underlying presumed consent law.

Though this work is a simple statistical analysis of comments and replies generated by a small number of online users, it is the first to examine online public commentary to the presumed consent law in Nova Scotia and can contribute to raise awareness about organ donation with the design of improved public outreach.

COMMENTS

(1) It seems not to be the case that Brazil implemented presumed consent legislation. Please, double check the sentence “Some countries, such as Singapore, Brazil, and Chile11,12,13 implemented presumed consent legislation but were unsuccessful at increasing donation and transplantation rates”

(2) What is the data collection period ? The dates of the two articles published on CBC (April 2nd and April 4th, 2019) are clearly stated but what is the period regarding all the 2663 respective comments and replies ?

(3) Is the code of program that collected the comments and replies available ? I would suggest the authors to provide more detail about this program and its workflow.

(4) I wound whether bots (fake accounts) played a role in “the negative commentary was greatly influenced by small group of dedicated commentators”.

Reviewer #2: This study has great relevance, especially when the information comes from social networks, which are gaining significant visibility. In addition, the public debate around organ donation is relevant and necessary as part of the guarantee of civil rights.

It would be great, in the introduction, to show the rates, in number, of organ donation from Nova Scotia and other Canadian provinces. Also, the name of the software that you use to collect the data. I did not fully understand the definition of positive, negative, and neutral comments in the methods section. Why did you collect the data in three days (April 2nd and April 4th, 2019)? Do you think that it might be a bias toward negative comments?

6. PLOS authors have the option to publish the peer review history of their article (what does this mean?). If published, this will include your full peer review and any attached files.

Reviewer #1: **Yes: **Diego Pinheiro

Reviewer #2: **Yes: **Sueny Paloma Lima dos Santos

---

## [Author Response · Author response to Decision Letter 0]

17 Feb 2022

Response to reviewer comments

Reviewer 1 comment: (1) It seems not to be the case that Brazil implemented presumed consent legislation. Please, double check the sentence “Some countries, such as Singapore, Brazil, and Chile11,12,13 implemented presumed consent legislation but were unsuccessful at increasing donation and transplantation rates”

Response: We’re a bit unsure of this critique. In this statement we wanted to state that in the past the countries listed implemented a version of deemed or presumed consent and were unsuccessful at raising rates. To our knowledge and as evidenced in our cited examples as well as other research, Brazil most certainly did implement presumed consent and faced significant backlash from the public after doing so. (For more examples please see: https://www.thelancet.com/journals/lancet/article/PIIS0140-6736(05)60767-2/fulltext ; or https://link.springer.com/article/10.1007/s10620-019-05483-z or https://www.ncbi.nlm.nih.gov/pmc/articles/PMC3363073/). We don’t think this is an opinion or a controversial interpretation, rather common knowledge. Please advise if our statement is incorrectly phrased. 

Reviewer 1 comment: (2) What is the data collection period? The dates of the two articles published on CBC (April 2nd and April 4th, 2019) are clearly stated but what is the period regarding all the 2663 respective comments and replies?

Response: We collected the data on May 24th 2019. We have included this statement in the methods. 

Reviewer 1 comment: (3) Is the code of program that collected the comments and replies available? I would suggest the authors to provide more detail about this program and its workflow.

Response: We are providing the git hub link to the program, which has been available to the public since it was created. We think our description adequately describes what the program does as it as a simple scraper. If anyone wishes to have further coding details, the link can be accessed.

Complete script available here: https://giters.com/philbot9/python-cbc-comment-scraper/watchers

We have also added the following line for additional clarity: “The complete data sets were organized in CSV files.” 

Reviewer 1 comment: (4) I wound whether bots (fake accounts) played a role in “the negative commentary was greatly influenced by small group of dedicated commentators”. 

Response: This was also something our team discussed when conducting the analysis, but conducting analysis on participant accounts fell outside of the purview of this project. We therefore did not try running one of the few bot detection programs that some researchers have developed. As mentioned earlier, to acknowledge the importance of this point, we have included the following line in our discussion:

“It is possible that automated accounts, or “bots,” were contributing to the negativity in the discussions, but analyzing user accounts fell outside of the purview of this research. Future research on the topic could make use of developing software to estimate bot activity in such discussions. Future research could also examine public discussions on other social media sites such as Facebook, Reddit, or Twitter to provide additional insights into the discussions taking place.”

Reviewer 2 comment (1): It would be great, in the introduction, to show the rates, in number, of organ donation from Nova Scotia and other Canadian provinces. Also, the name of the software that you use to collect the data.

Response: We do not feel that including those specific numbers would add consider value and that including them would clutter the introduction. Those wishing to see the numbers can easily access the data in the reference we have provided, which links to an open-access database. If the editors of the journal believe it is good idea to include these numbers, we would be happy to include them. 

We have provided a link to the software we developed to download comments from the CBC’s website. 

Reviewer 2 comment (2): I did not fully understand the definition of positive, negative, and neutral comments in the methods section.

Response: We agree with this comment and have added the following to the Methods section, which should provide additional clarity: 

“Comments supporting and promoting the new legislation as well as donation and transplantation broadly were deemed positive whereas commenting critiquing the new legislation as well as donation and transplantation broadly were deemed negative. Comments which did not exhibit negativity or positivity, or which did not present a clearly defined negative or positive position, were coded as neutral.”

Reviewer 2 Comment (3): Why did you collect the data in three days (April 2nd and April 4th, 2019)? Do you think that it might be a bias toward negative comments?

Response: We appreciate this inquiry, but we do not think our data collection procedures would indicate any bias towards negative comments. We provided ample time for discussion to take place. If anything, more negativity might have been removed from the discussions as moderators would likely have blocked and removed exceedingly aggressive, offensive, or hateful comments based on their criteria for doing so.

---

## [Decision Letter · Decision Letter 1]

20 Sep 2022

PONE-D-21-22056R1Analyzing online public commentary responding to the announcement of deemed consent organ donation legislation in the Canadian province of Nova ScotiaPLOS ONE

Dear Dr. Marcon,

Thank you for submitting your manuscript to PLOS ONE. After careful consideration, we feel that it has merit but does not fully meet PLOS ONE’s publication criteria as it currently stands. Therefore, we invite you to submit a revised version of the manuscript that addresses the points raised during the review process. We have obtained additional reviews from reviewers with relevant methodological expertise. Please address all reviewer requests. In addition, please also address the following request: Thank you for addressing in the discussion section the possibility of bots affecting your results. It is not clear however that assessing this possible effect is outside the purview of this research. Please further discuss, in the discussion section, this possibility as a limitation and provide plausible estimates to quantify this effect if possible.

We look forward to receiving your revised manuscript.

Kind regards,

Yann Benetreau, PhD

Division Editor

PLOS ONE

Reviewers' comments:

Reviewer's Responses to Questions

**Comments to the Author**

1. If the authors have adequately addressed your comments raised in a previous round of review and you feel that this manuscript is now acceptable for publication, you may indicate that here to bypass the “Comments to the Author” section, enter your conflict of interest statement in the “Confidential to Editor” section, and submit your "Accept" recommendation.

Reviewer #2: All comments have been addressed

Reviewer #3: (No Response)

Reviewer #4: (No Response)

Reviewer #5: (No Response)

2. Is the manuscript technically sound, and do the data support the conclusions?

Reviewer #2: Yes

Reviewer #3: Partly

Reviewer #4: Yes

Reviewer #5: Partly

3. Has the statistical analysis been performed appropriately and rigorously? 

Reviewer #2: Yes

Reviewer #3: No

Reviewer #4: Yes

Reviewer #5: Yes

4. Have the authors made all data underlying the findings in their manuscript fully available?

Reviewer #2: Yes

Reviewer #3: No

Reviewer #4: Yes

Reviewer #5: Yes

5. Is the manuscript presented in an intelligible fashion and written in standard English?

Reviewer #2: Yes

Reviewer #3: Yes

Reviewer #4: Yes

Reviewer #5: Yes

6. Review Comments to the Author

Reviewer #2: All comments have been addressed and I believe that there is no dual publication, or research ethics problem.

Reviewer #3: Comment 1: the link of the does not shows any information (i.e repository is not activate) webaddress https://giters.com/philbot9/python-cbc-commentscraper/watchers (either remove it or make activate it)

Comment 2: Table captions usually will be kept above the table. In table 1 both above and below the table caption was kept. Remove the caption from the bottom of the table.

Comment 3: While using ANOVA they should have to follow certain assumptions. (see the web address for assumptions of ANOVA How to Check ANOVA Assumptions - Statology) and the data should be of type either of Interval or ratio scale.

Comment 4: Since the authors has written “The complete data sets were organized in CSV files”, can you make available the CSV file? Similarly, the authors have quoted that “All coding analysis was conducted in shared Google Sheets”, so, if these sheets are made available then the authenticity of the research will be increased.

Comment 5: on the analysis part “ Article 1 demonstrated a higher percentage of negativity in comments (n = 251, 56.0%) than Article 2 (n = 88, 48.6%), but the difference was not significant, (X2 = 3.08, df = 2, p = .21).” I don't think the chi square value will be so less (3.08) as the difference (251-88) is so high (there must be a significant difference with chi square value more than 50.

Comment 6: If the data follows all the assumptions of ANOVA, then also the analysis done to find F value is not consistent. So I would like to request you the syntax of the analysis along with data to verify and clarify the problem.

Comment 7: Outlook of table is not in the scientific table format. Please arrange it.

Comment 8: Requested to include DOI for References wherever possible.

Reviewer #4: Comment 1: As per your reply to Reviewer 2 Comment (3), it seems that you have collected the data in three days (April 2nd and April 4th, 2019). However, in the methodology, you wrote "On May 24th, 2019, we downloaded all user comments and replies..". Does it mean that you downloaded all the comments in those two articles made since the date it was published and till March 24, or you just downloaded the comments between April 2nd and April 4th? This could be more clarified in the methodology section.

Reviewer #5: The authors perform a content analysis of comment sections in two Canadian Broadcasting Company articles, which announce changes to Nova Scotia’s deemed consent organ donation. In general, their findings suggest that negative-leaning comments comprised the majority of comments and received more replies than negative comments. Further, they noted that negative comments were most associated with “human rights,” “body autonomy,” and legality with present legislation. Positive and neutral-leaning comments shared themes of “altruism,” “ body autonomy,” and “effectiveness.” The authors provide some suggestions for the government and stakeholders to “express why [there are] changes around consenting” and “how policy will be enacted,” which seem to be helpful for the community. The analyses, as stated by the authors, are limited by the lack of demographic data on commenters and the lack of exploring the possibility of bot-generated comments. Unfortunately, the conclusions are weakened by the ambiguity of hypotheses and lack of addressing previous comments on the background of the research.

Please find detailed comments below:

Major:

Introduction: The hypothesis, “We hypothesized the existence of diverse and strong opinions that may potentially have influenced public perception and debate on this topic” does not seem to be addressed by your selected methods of analysis. The descriptor, “diverse” may denote a demographic category that, as mentioned in the Methods & Discussion, was not considered in your analysis. Furthermore, there is no qualification within the content categories for what a “strong” opinion would be. This, in turn, makes the conclusions relating to positive and neutral comments appear not to address whether or not there are opinions that influence public perception and debate. Perhaps, the authors could revise this hypothesis by replacing “diverse and strong opinions” with descriptors that more clearly outline the possible theory upon which their inductive approach may arrive at.

Minor:

Introduction, paragraph 1: The authors chose to not address previous suggestions to provide quantification on “relatively high rates of organ donation compared to other Canadian provinces.” These numbers would provide a clear perspective and improve readers’ overall understanding of the background.

Data Analysis: The link to the code does not appear to arrive at the correct GitHub repository.

Discussion, paragraph 2: Authors’ use of qualifiers such as “heated” and “relatively large” are not supported by the results of the analyses.

Discussion, paragraph 3: While the addition of mention of bots or lack of demographic information on commenters is mentioned, there is no mention of the weaknesses of this analysis and how that affects the recommendations the authors give to the community. Morse and Richards, 2002 and Polit and Beck 2006 provide insight into why this can be problematic.

7. PLOS authors have the option to publish the peer review history of their article (what does this mean?). If published, this will include your full peer review and any attached files.

Reviewer #2: **Yes: **Sueny Paloma Lima dos Santos

Reviewer #3: **Yes: **Bijay Lal Pradhan

Reviewer #4: No

Reviewer #5: No

---

## [Author Response · Author response to Decision Letter 1]

3 Nov 2022

Response to reviewers 

Reviewer #3: Comment 1: the link of the does not shows any information (i.e repository is not activate) web address https://giters.com/philbot9/python-cbc-commentscraper/watchers (either remove it or make activate it)

Response: This has been corrected. We corrected a typo in the link. 

Comment 2: Table captions usually will be kept above the table. In table 1 both above and below the table caption was kept. Remove the caption from the bottom of the table.

Response: Thank you. This is done. 

Comment 3: While using ANOVA they should have to follow certain assumptions. (see the web address for assumptions of ANOVA How to Check ANOVA Assumptions - Statology) and the data should be of type either of Interval or ratio scale.

Response: Thank you for this suggestion. Our data did not meet the normal distribution assumption, so we ran a Kruskal-Wallis test with a post-hoc Dunn test instead. This did not alter our findings drastically and we have updated the information in the manuscript.

Comment 4: Since the authors has written “The complete data sets were organized in CSV files”, can you make available the CSV file? Similarly, the authors have quoted that “All coding analysis was conducted in shared Google Sheets”, so, if these sheets are made available then the authenticity of the research will be increased.

Response: The editors at PlosOne informed us that we were required to make all original data available as part of the submission. We had a conflict because our data contained the usernames of participants in discussions. To make the data available while still protecting the privacy of the users, we created an anonymized dataset and made this data available. It will be available with the publication. Having the complete data set plus the coding frame allows the study to be replicated. 

Comment 5: on the analysis part “ Article 1 demonstrated a higher percentage of negativity in comments (n = 251, 56.0%) than Article 2 (n = 88, 48.6%), but the difference was not significant, (X2 = 3.08, df = 2, p = .21).” I don't think the chi square value will be so less (3.08) as the difference (251-88) is so high (there must be a significant difference with chi square value more than 50.

Response: The chi square test compared the articles with the tone of comments (i.e. negative, neutral, positive) not just negative comments. We have revised the statement about significance to clarify what the data is in reference to.

Comment 6: If the data follows all the assumptions of ANOVA, then also the analysis done to find F value is not consistent. So I would like to request you the syntax of the analysis along with data to verify and clarify the problem.

Response: Please see our response to comment #3

Comment 7: Outlook of table is not in the scientific table format. Please arrange it.

Response: We have included completed borders for this table. We will make additional formatting changes to these tables if the editors request it. 

Comment 8: Requested to include DOI for References wherever possible.

Response: We will be happy to follow whatever formatting requests the editors of Plos One have for the text as well as for the reference list. We will follow their guidelines.

Reviewer #4: Comment 1: As per your reply to Reviewer 2 Comment (3), it seems that you have collected the data in three days (April 2nd and April 4th, 2019). However, in the methodology, you wrote "On May 24th, 2019, we downloaded all user comments and replies..". Does it mean that you downloaded all the comments in those two articles made since the date it was published and till March 24, or you just downloaded the comments between April 2nd and April 4th? This could be more clarified in the methodology section.

Response: We have clarified this section. It now reads, “We therefore downloaded all comments which had been left in discussions, and not removed by moderators, between the publication dates and May 24th.” 

Reviewer #5: 

Major:

Introduction: The hypothesis, “We hypothesized the existence of diverse and strong opinions that may potentially have influenced public perception and debate on this topic” does not seem to be addressed by your selected methods of analysis. The descriptor, “diverse” may denote a demographic category that, as mentioned in the Methods & Discussion, was not considered in your analysis. Furthermore, there is no qualification within the content categories for what a “strong” opinion would be. This, in turn, makes the conclusions relating to positive and neutral comments appear not to address whether or not there are opinions that influence public perception and debate. Perhaps, the authors could revise this hypothesis by replacing “diverse and strong opinions” with descriptors that more clearly outline the possible theory upon which their inductive approach may arrive at.

Response: We have modified this section by removing the adjectives “diverse” and “strong” and replaced them with “positive” and “negative.” 

Minor:

Introduction, paragraph 1: The authors chose to not address previous suggestions to provide quantification on “relatively high rates of organ donation compared to other Canadian provinces.” These numbers would provide a clear perspective and improve readers’ overall understanding of the background.

Response: We thank you for this request. We re-examined the data, and realized we had not correctly summarize the report. We have modified the manuscript accordingly, and it now reads, “In 2019, Atlantic Canada had lower rates of organ donation (13/1 million) compared to other Canadian provinces (Canada average of 21/ 1 million), while internationally, Canada’s organ donation rates were lower than many other national averages.”

Data Analysis: The link to the code does not appear to arrive at the correct GitHub repository.

Response: Thank you for catching this. There was a typo in the original link, and it has been corrected. 

Discussion, paragraph 2: Authors’ use of qualifiers such as “heated” and “relatively large” are not supported by the results of the analyses.

Response: We agree with this critique. These adjective have been removed. 

Discussion, paragraph 3: While the addition of mention of bots or lack of demographic information on commenters is mentioned, there is no mention of the weaknesses of this analysis and how that affects the recommendations the authors give to the community. Morse and Richards, 2002 and Polit and Beck 2006 provide insight into why this can be problematic.

Response: We appreciate this feedback but do not feel that an inclusion of this literature strengthens our straightforward study. If the editors of the journal would like use to expand the limitations section, we would be happy to do so.

---

## [Decision Letter · Decision Letter 2]

29 Nov 2022

Analyzing online public commentary responding to the announcement of deemed consent organ donation legislation in the Canadian province of Nova Scotia

PONE-D-21-22056R2

Dear Dr. Marcon,

We’re pleased to inform you that your manuscript has been judged scientifically suitable for publication and will be formally accepted for publication once it meets all outstanding technical requirements.

Kind regards,

Ali B. Mahmoud, Ph.D.

Academic Editor

PLOS ONE

Additional Editor Comments (optional):

Reviewers' comments:

Reviewer's Responses to Questions

**Comments to the Author**

1. If the authors have adequately addressed your comments raised in a previous round of review and you feel that this manuscript is now acceptable for publication, you may indicate that here to bypass the “Comments to the Author” section, enter your conflict of interest statement in the “Confidential to Editor” section, and submit your "Accept" recommendation.

Reviewer #2: All comments have been addressed

Reviewer #3: All comments have been addressed

Reviewer #4: All comments have been addressed

2. Is the manuscript technically sound, and do the data support the conclusions?

Reviewer #2: Yes

Reviewer #3: Yes

Reviewer #4: Yes

3. Has the statistical analysis been performed appropriately and rigorously? 

Reviewer #2: Yes

Reviewer #3: Yes

Reviewer #4: Yes

4. Have the authors made all data underlying the findings in their manuscript fully available?

Reviewer #2: Yes

Reviewer #3: Yes

Reviewer #4: Yes

5. Is the manuscript presented in an intelligible fashion and written in standard English?

Reviewer #2: Yes

Reviewer #3: Yes

Reviewer #4: Yes

6. Review Comments to the Author

Reviewer #2: The manuscript is completed successfully, and each and every remark has been taken into consideration. Great work.

Reviewer #3: Since data did not follow normality Kruskal-Wallis test with pairwise post-hoc Dunn test that you have used is good. All the comments has been addressed.

Reviewer #4: (No Response)

7. PLOS authors have the option to publish the peer review history of their article (what does this mean?). If published, this will include your full peer review and any attached files.

Reviewer #2: No

Reviewer #3: **Yes: **Bijay Lal Pradhan

Reviewer #4: No

---

## [Editor Report · Acceptance letter]

5 Dec 2022

PONE-D-21-22056R2 

Analyzing online public commentary responding to the announcement of deemed consent organ donation legislation in the Canadian province of Nova Scotia 

Dear Dr. Marcon:

I'm pleased to inform you that your manuscript has been deemed suitable for publication in PLOS ONE. Congratulations! Your manuscript is now with our production department. 

Kind regards, 

on behalf of

Dr. Ali B. Mahmoud 

Academic Editor

PLOS ONE